# Autoimmune Effect of Antibodies against the SARS-CoV-2 Nucleoprotein

**DOI:** 10.3390/v14061141

**Published:** 2022-05-25

**Authors:** Daria Matyushkina, Varvara Shokina, Polina Tikhonova, Valentin Manuvera, Dmitry Shirokov, Daria Kharlampieva, Vasily Lazarev, Anna Varizhuk, Tatiana Vedekhina, Alexander Pavlenko, Leonid Penkin, Georgij Arapidi, Konstantin Pavlov, Dmitry Pushkar, Konstantin Kolontarev, Alexander Rumyantsev, Sergey Rumyantsev, Lyubov Rychkova, Vadim Govorun

**Affiliations:** 1Scientific Research Institute for Systems Biology and Medicine, Scientific Driveway, 18, 117246 Moscow, Russia; varvaramys@gmail.com (V.S.); pavav@mail.ru (A.P.); leopold.valerjanovitch@yandex.ru (L.P.); vgovorun@yandex.ru (V.G.); 2Federal Research and Clinical Center of Physical-Chemical Medicine, Malaya Pirogovskaya, 1a, 119435 Moscow, Russia; tikhonova.polly@mail.ru (P.T.); vmanuvera@yandex.ru (V.M.); dmitry.a.shirokov@gmail.com (D.S.); harlampieva_d@mail.ru (D.K.); lazar0@mail.ru (V.L.); aliviense@gmail.com (A.V.); neglect1@yandex.ru (T.V.); arapidi@gmail.com (G.A.); qpavlov@mail.ru (K.P.); 3Bioinformatics and Genomics Intercollege Graduate Program, Huck Institutes of Life Sciences, The Pennsylvania State University, University Park, PA 16802, USA; 4Moscow Institute of Physics and Technology, National Research University, 117303 Dolgoprudny, Russia; 5City Clinical Hospital named after S.I. Spasokukotsky Department of Health of City of Moscow, Vuchetich str., 21, 127206 Moscow, Russia; pushkardm@mail.ru (D.P.); kb80@yandex.ru (K.K.); 6Hospital of the Russian Academy of Sciences, Oktyabrsky Prospect, 3, 108840 Troitsk, Russia; alexrum47@mail.ru (A.R.); s_roumiantsev@mail.ru (S.R.); 7Scientific Centre for Family Health and Human Reproduction Problems, 16 Timiryazev str., 664003 Irkutsk, Russia; rychkova.nc@gmail.com

**Keywords:** COVID-19, coronavirus infection, SARS-CoV-2, nucleoprotein, antibodies, autoimmunity

## Abstract

COVID-19 caused by SARS-CoV-2 is continuing to spread around the world and drastically affect our daily life. New strains appear, and the severity of the course of the disease itself seems to be decreasing, but even people who have been ill on an outpatient basis suffer post-COVID consequences. Partly, it is associated with the autoimmune reactions, so debates about the development of new vaccines and the need for vaccination/revaccination continue. In this study we performed an analysis of the antibody response of patients with COVID-19 to linear and conformational epitopes of viral proteins using ELISA, chip array and western blot with analysis of correlations between antibody titer, disease severity, and complications. We have shown that the presence of IgG antibodies to the nucleoprotein can deteriorate the course of the disease, induce multiple direct COVID-19 symptoms, and contribute to long-term post-covid symptoms. We analyzed the cross reactivity of antibodies to SARS-CoV-2 with own human proteins and showed that antibodies to the nucleocapsid protein can bind to human proteins. In accordance with the possibility of HLA presentation, the main possible targets of the autoantibodies were identified. People with HLA alleles A01:01; A26:01; B39:01; B15:01 are most susceptible to the development of autoimmune processes after COVID-19.

## 1. Introduction

People around the world commonly get infected with human coronaviruses 229E, NL63, OC43, and HKU1, which generally cause mild to moderate upper-respiratory tract illness, presumably contributing to 15–30% of cases of common colds in humans [1]. Sometimes, coronaviruses that infect animals can evolve and make people sick and become a new human coronavirus. Among them, SARS-CoV and MERS-CoV caused outbreaks in 2002 [2] and 2012 [3], respectively. SARS-CoV-2 (COVID-19) is the most recently discovered. It first occurred in Wuhan, China in December 2019, and it swiftly spread across China and has been aggressively infecting people globally. The Severe Acute Respiratory Distress Syndrome (SARS) and Middle East Respiratory Syndrome (MERS) cases were reported to have a very high case fatality rate of 9.5 and 34.4%, respectively. In contrast, COVID-19 has a case fatality rate of 2.13% [4]. In addition, a significant percentage of people carry this infection in a mild or generally asymptomatic form, but despite this, compared with the usual flu, the recovery period after suffering COVID-19 is longer and sometimes accompanied by varied longer-term sequelae.

Humoral immunity is one of the key lines of defense against viral infection. As has been shown in many studies, antibodies to both structural and non-structural proteins of SARS-CoV-2 virus are detected in the sera of individuals having recovered from COVID-19 [5,6]. In the asymptomatic group of patients, the virus-specific IgG levels are significantly lower relative to the symptomatic group in the acute phase [7]. Several research groups associate the development of a severe course of the disease and post-covid syndrome with the development of autoimmune processes caused by antibodies against SARS-CoV-2 [8,9] or with the presence of pre-existing autoantibodies in the body [10,11]. 

It is clear that most people infected with SARS- CoV-2 display an antibody response between 10 and 14 days after infection. The nucleocapsid and spike antigens are most frequently used in diagnostic analysis. Spike (S) glycoprotein facilitates entry into the host cells and is the main target of neutralizing antibodies. Furthermore, new data indicate alternative ways of adhesion of the virus [12,13]; therefore, the presence of antibodies to the S protein does not necessarily provide absolute protection against the virus invasion. Nevertheless, most of the convalescent plasma samples obtained from individuals having recovered from COVID-19 did not have high levels of neutralizing activity [14]. Thus, in order to understand how to choose a correct tactic for creating an effective and harmless vaccine for humans, we should first understand how the immune system responds to the coronavirus SARS-CoV-2.

Moreover, despite the direct effect of virus proteins on the host cell, there are discussions about the possible cross-reactivity of antibodies against SARS-CoV-2 with the human’s own proteins, which can significantly aggravate the disease progression [15,16]. It is known that one of the factors causing autoimmune diseases is infections caused by viruses, such as the Coxsackie B virus (possible involvement in the occurrence of diabetes mellitus) [17], hepatitis C virus and Dengue virus (associated with systemic lupus erythematosus) [18,19], as well as herpesviruses and measles viruses (associated with multiple sclerosis) [20,21]. There are cases when autoimmune diseases, such as Guillain-Barré syndrome or systemic lupus erythematosus, develop after COVID-19 infection [22]. It is speculated that SARS-CoV-2 can disturb self-tolerance and trigger autoimmune responses through cross-reactivity with host cells. It was found that 20% of people hospitalized with severe COVID-19 had high or intermediate levels of autoantibodies to type I IFNs. Autoantibodies were also found in at least 18% of people who died from the disease. In contrast, people with no or mild symptoms had very low levels of these autoantibodies. The researchers estimated that the autoantibodies may account for about 20% of total fatal COVID-19 cases [23]. The mechanisms of such autoantibody production are not yet clear. Prolonged inflammation during severe COVID-19 may cause the immune system to produce antibodies to different viral proteins (structural and nonstructural). The amino acid sequences of such proteins can overlap with human proteins, and it could trigger the production of autoantibodies. Furthermore, evidence suggests that some medications that suppress the immune system appear to improve survival in severe COVID-19 [8].

Therefore, in this work, we performed a large-scale analysis of the humoral response of patients with COVID-19 to structural and non-structural proteins of SARS-CoV-2 with the assessment of its correlation with symptoms, and we attempted to identify the possible cross-reactivity of antibodies against the virus proteins with human proteins. 

## 2. Materials and Methods

### 2.1. Sample Information

This study enrolled a total of 149 patients from several public health clinical centers who had recovered from COVID-19. All the patients had COVID-19 diagnoses confirmed by RT-PCR and/or computed tomography (CT) clinical evidence accompanied by the presentation of medium or severe symptoms. The median age of patients was 59 years; 47% of the patients were male. More detailed comparative characteristics of patients’ indicators are presented in Appendix A. The analysis was carried out for sera obtained at different stages of the disease progression: at the time of admission to the hospital (S1), after 48 h of hospitalization (S2), after 7 days (S3), and at the time of discharge from the hospital (S7). The number of days between the disease inception and hospitalization was also considered. The number of illness days was counted from the first occurrence of any (SARS) symptoms according to the patient accounts. For the analyses, patients were grouped into three categories according to radiographic data (CT): mild severity patients with grades between 0–1.5, medium severity patients with grades 2 and 2.5, and high severity patients with grades from 3 to 4. The basis of the CT score is as follows: CT = 0, the norm and absence of CT signs of viral pneumonia against the background of a typical clinical picture and a relevant epidemiological anamnesis; CT = 1, frosted glass sealing zones, lung parenchymal involvement ≤25% into the pathological process or absence of CT signs of viral pneumonia against the background of a typical clinical picture and a relevant epidemiological anamnesis; CT = 2, frosted glass sealing zones, involvement of the lung parenchyma 25–50% into the pathological process; CT = 3, frosted glass sealing zones, consolidation zones, involvement of the lung parenchyma 50–75% into the pathological process; CT = 4, diffuse compaction of lung tissue of the ground glass type and consolidation in combination with reticular changes, hydrothorax (bilateral, predominantly on the left), involvement of the lung parenchyma ≥75% into the pathological process. Sera from 48 healthy people collected in the before-COVID-19 era were used as controls. All human donors voluntarily gave informed consent prior to being enrolled in the study. 

### 2.2. Eukaryotic Cells Culture

Human cell lines A549 and T-84 were obtained from the ATCC. A549 was grown in Dulbecco modified Eagle medium (Invitrogen, Carlsbad, CA, USA) containing 10% fetal bovine serum (Corning Cellgro, Carlsbad, CA, USA). T-84 was grown in a 1:1 mixture of Ham F12 and DMEM maintenance medium supplemented with 10% FCS. In both cases, media contained penicillin-streptomycin (Invitrogen, Carlsbad, CA, USA), 0.1 mM non-essential amino acids (Invitrogen, Carlsbad, CA, USA), and 1.0 mM sodium pyruvate (Invitrogen, Carlsbad, CA, USA). All cell lines were maintained and incubated at 37 °C in atmosphere containing 5% CO_2_.

### 2.3. Obtaining Recombinant SARS-CoV-2 Proteins

Human codon optimized, 2xStrep-tagged SARS-CoV-2 ORFs containing plasmids were kindly provided by David Gordon [24]. The human cell line Expi293F constantly expressing SARS-CoV-2 ORFs was generated using an Expi29 Expression System Kit (Life Technologies, Carlsbad, CA, USA). Geneticin (500 µg/mL) was added 48 h post transfection and G-418-resistant lines were propagated. The selection of target proteins was as described earlier [25]. 

### 2.4. ELISA and CMIA

The level of IgG antibodies in the sera of patients with COVID-19 was determined using an enzyme-linked immunosorbent assay (ELISA) «SARS-CoV-2 IgG» (kits supplied by “Lytech” Co. Ltd. (Moscow, Russia)) (S2 and nucleoprotein) and «SARS-CoV-2-IgG-RBD-ELISA» (kits supplied by «The center of innovation of biotechnology Allele», Novosibirsk, Russia) according to the manufacturer’s instructions. Serum dilution was 1:10. Sample wells were washed five times with a wash buffer. After washing, 100 µL conjugate was added and incubated at 37 °C for 30 min. The TMB reagent was added in volume 100 µL and incubated at 37 °C for 10 min in the dark. After that, 100 µL of the stop-solution was added in each well, and absorbance of the sample wells measured immediately at 450 nm and 620–650 nm for both ELISA methods OD_critical_ = OD_mean_ K- + 0.2, where OD_mean_ K- is the average OD K- for two negative control wells. The result of the analysis was quantified by a coefficient of positivity: CP = OD_sample_/OD_critical_. The result of the analysis was considered positive if the CP was more than 1.1 and negative if the CP was less than 0.9; a CP within 0.9–1.1 was considered equivocal. 

The presence of antibodies to RBD was confirmed by an alternative method. The SARS-CoV-2 IgG II Quant assay (Abbott, Dublin, Ireland) as described in [26] was used. The result of the analysis was considered positive if the value (BAU/mL) was more than or equal to 50.

### 2.5. Western Blot Analysis

Recombinant SARS-CoV-2 proteins were fractionated by 1D SDS PAGE with Laemmli sample buffer (BioRad, Hercules, CA, USA) as a denaturing condition. Then, 4–10% PAGE gels were transferred to a PVDF membrane (GE Healthcare, Chicago, IL, USA) preactivated with 100% methanol under an alternating voltage of 100 V amplitude. The membrane was blocked with PBS containing 0.05% Tween-20 and 3% skim milk (Blotting-Grade Blocker, Bio-Rad, Hercules, CA, USA) at 4 °C overnight. Then, sera were used in 1:50 dilution and incubated at RT for 1 h. After washing the membrane, the protein-bound antibodies were stained with 1:50000 diluted HRP-Conjugated goat anti-human IgG (Millipore, Burlington, MA, USA) and incubated at RT for 1 h. Protein bands were detected using ECL Plus Western Blotting Detection System (GE Healthcare) and iBright Imaging Systems (ThermoFisher, Waltham, MA, USA). Competitive western blot analysis was performed using the preincubation of the analyzed serum with 30 µg of recombinant NP in conformational and linear forms. Quantitative values of chemiluminescence were calculated using the built-in software.

### 2.6. Trypsin Digestion and MS Analysis

The protein bands after 1D-PAGE were subjected to trypsin in-gel hydrolysis with subsequent MALDI analysis as described [27]. 

### 2.7. Array Fabrication

The arrays were fabricated on 75*25*1-mm Pyrex glass slides (Corning, New York, NY, USA). On each slide, 5*5-mm square zones were marked using a CNC CO_2_ Laser cutting machine GLS Hybrid 100/50 (settings: 20 W, 1500 pulses per inch). The marked slides were polished chemically by immersing them into the H_2_SO_4_/H_2_O_2_ mixture, then they were rinsed multiple times with distilled water and dried at 80 °C. Protocols for the modification of the slide surface with 3-aminopropyltriethoxysilane (APTES) and its functionalization for protein immobilization were developed based on the previously published general procedures [28]. All reagents were purchased from Sigma-Aldrich (St. Louis, MO, USA). SARS-CoV-2 recombinant proteins or fibrinogen from human plasma (Sigma-Aldrich, St. Louis, MO, USA) were diluted in a 10 mM potassium-phosphate buffer (pH 7.4) to a concentration of 0.1 mg/mL, and the solutions were applied onto the slide surface using the ITWO-300P piezo driven micro-dispenser (M2-Automation Systems, Berlin, Germany) following the predesigned scheme (Appendix A). Free surface of the arrays was blocked by BSA. Arrays were scanned using a Nikon Eclipse Ti2 microscope (Nikon, Tokyo, Japan) with a DS-Qi2 high-speed camera with a 90 msec acquisition time. The resulting images were analyzed using ImageJ version 1.53i software (National Institutes of Health, Bethesda, MD, USA). The fluorescence signal in each spot (F) was calculated as the average apparent fluorescence minus the average background. For each protein, two replicates were averaged, and a normalized signal was calculated as F/Fo ± SD, where Fo was the average buffer signal. Normalized array signals (F/Fo) indicating relative antibody levels in plasma samples showed a skewed distribution in both COVID-positive and control groups. Therefore, the statistical significance of the difference between these groups was verified using the Mann-Whitney U-test. Briefly, for each protein, normalized signal values in COVID-positive and control groups were ranked and compared pairwise, which provided U_COVID_ and U_control_ values (the total number of times F_COVID_/Fo > F_control_/Fo, and vice versa). Then, U was calculated as min (U_COVID_, U_control_). Because the total sample size (N = n_COVID_ + n_control_) was large enough (n_COVID_*n_control_ > 20), normal approximation with the respective z-score equation was used $\frac{U-\frac{n^{\mathrm{COVID}}\cdot n^{\mathrm{control}}}{2}}{\sqrt{\frac{n^{\mathrm{COVID}}\cdot n^{\mathrm{control}}\left(N+1\right)}{12}}\;}$.

### 2.8. Bioinformatics Analysis of Amino Acid Cross-Reactivity between Cell Lines and SARS-CoV-2 S and N Proteins

For the analysis of common regions between N, S SARS-CoV-2, and human proteins, all protein subsequences of length 6 between A549 [29] and T84 [30] cell line proteins, which could be found in SARS-CoV-2 proteins [31] N and S, were computed. For depicting target proteins (which were detected by mass-spectrometry), the computation process was the same, except for the length of the intersected subsequences. The subsequences shared by SARS-CoV-2 proteins of length 4 were computed. The shared subsequences of length 6 or 4 for A549 and T84 cell line proteins and target proteins, respectively, and structural information of N [32] and S [33] proteins were also analysed. The mutation profile was computed using sliding windows with a width of 20 amino acids based on the CoV-GLUE database information [34].

### 2.9. Analysis of Peptide Binding to HLA Class I and II

We used the downloaded standalone version of NetMHCpan 4.1 and NetMHCIIpan 4.0 [35] to predict protein fragments likely binding the most common alleles of HLA class I and the pan-specific binding of peptides to HLA class II alleles of all known sequences, respectively. The analysis parameters included “fasta” input type. According to the developer’s recommendation, we chose HLA-A01:01, HLA-A02:01, HLA-A03:01, HLA-A24:02, HLA-A26:01, HLA-B07:02, HLA-B08:01, HLA-B27:05, HLA-B39:01, HLA-B40:01, HLA-B58:01, and HLA-B15:01 as the HLA class I supertype representative alleles. For NetMHCpan, we chose the threshold for strong and weak binders as 0.5% and 2% rank, respectively. The rank was evaluated based on artificial neural networks, trained on an extensive dataset of Eluted Ligand (EL) mass spectrometry measurements, covering the three human HLA class I isotypes HLA-A, HLA-B, and HLA-C. For NetMHCIIpan, we chose the threshold for strong and weak binders as 2% and 10% rank, respectively. The rank was evaluated based on neuron architecture, trained on an extensive dataset of Eluted Ligand (EL) mass spectrometry measurements, covering the three human HLA class II isotypes HLA-DR, HLA-DQ, and HLA-DP.

### 2.10. Statistics

All pictures and data analyses were produced with the Python version 3.7.7 (Centrum voor Wiskunde en Informatica Amsterdam, The Netherlands) and SciPy version 1.5.0 stats module [36]. The comparisons of ELISA values between groups, using/non-using Oxygen tools, were performed using an ANOVA test (f_oneway function). The sample size was large enough (~330 samples) to detect significance around a small effect size (0.1), considering the statistical power as 0.8 at a 0.05 significance level [37]. The comparison between the days of appearance of the antibody response according to the western blot and cross-reactivity was performed using non-parametric tests (Mann-Whitney and Fisher exact test from the SciPy stats module, correspondingly) due to the small sample size. 

Mass-spectrometry analysis was performed in four biorepeats. The statistical processing of data obtained using Western blot analysis was performed by the built-in software iBright Imaging Systems (ThermoFisher).

## 3. Results

### 3.1. Humoral Response Dynamics Following Infection with SARS-CoV-2

Understanding the antibody dynamics following SARS-CoV-2 infection is essential for associating the antibody titer with disease severity. Antibodies of conformational epitopes of recombinant nucleoprotein (NP), the receptor-binding-domain (RBD) and S2 domain of the S-protein, nsp2, nsp5, nsp7, nsp9, nsp10, and nsp15 were analyzed using ELISA and chip array. For part of the ELISA RBD, data were verified by CMIA (Appendix A). Antibodies of linear epitopes were analyzed by the western blot. The antibody level was deemed high if it was above the first quartile of the antibody level distribution consisting of the first three timepoints. There were no statistically significant differences between samples from different hospitals or from patients of different ages and genders.

The timing of appearance of the antibodies of conformational epitopes to the NP, RBD, and S2 for the medium- and high-severity groups of patients was the same; all antibody values became positive within 14–16 days from the inception of the disease. Significant differences in antibody titer between the different severity groups of patients appeared after the 16th day of the disease (Appendix A), and all antibody tests also became positive after this period. Patients with a mild course of the disease had generally lower antibody levels than the medium- and high-severity groups of patients (Figure 1). The most significant differences in antibody titer were observed for the nucleocapsid protein. In the group of high severity patients, the level of antibodies was 2.5-fold higher (Figure 1a and Appendix A). 

We also analyzed the correlations between the antibodies of conformational epitopes and the different clinical parameters of patients (gender, age, temperature, oxygen demand, obesity, arterial hypertension, coronary heart disease, chronic heart failure, diabetes mellitus, smoking, chronic obstructive pulmonary disease, bronchial asthma, rheumatoid arthritis, inflammatory bowel disease, and liver cirrhosis). There was a significant correlation with levels of antibodies to NP and RBD only for oxygen demand, and for S2, it was not statistically significant (Figure 2 and Appendix A).

Humoral immune responses to the conformational SARS-CoV-2 protein epitopes were additionally profiled using protein microarrays. The arrays were designed to incorporate both structural (NP, RBD, and S2) and non-structural (NSP2, NSP5, NSP7, NSP9, NSP10, and NSP15) SARS-CoV-2 proteins. The NP was obtained in two variants (N* and N for Expi293 and *E. coli*-derived samples, respectively) to analyze the importance of post-translational modifications for antibody recognition. Human fibrinogen was used as a negative control. 

The protein arrays were probed with the sera of COVID-19-positive and negative (control) donors, and then they were rinsed and stained with Alexa546-labeled secondary antibodies. Array design and representative images obtained using fluorescence microscopy are shown in Appendix A. In order to verify reproducibility, three arrays were prepared independently and probed with the serum of the same patient. Variations between those biological repeats were comparable to variations between the technical repeats. Long-term storage had a substantial impact on the quality of the array analysis (Appendix A); thus, all the arrays were probed within two days after fabrication.

Major results of the protein array analysis are shown in Figure 3. The results accord qualitatively with those obtained by the ELISA assay. A particularly pronounced difference between COVID-19-positive donors (average: 18.9; median: 13,4) and the control (average: 2.3; median: 1.6) groups was observed in responses to the Expi293-derived nucleocapsid protein (N*). For non-structural proteins, the difference between the COVID-positive and control groups was statistically insignificant in all cases, but we obtained an intriguing outlier for the protease variant NSP5. 

Antibodies to linear epitopes of the same proteins appear 1–2 days later. However, not all patients had antibodies to the linear epitopes of N and S proteins (Figure 4a,b). Only NP had the same percentage of antibody detection as in ELISA (82%), while for RBD it was 17.5%, and for S2 it was 44.3%. Differences in the presence of antibodies to linear epitopes between the groups of patients was observed only for RBD (Figure 4c,d). In the group of patients with CT 0–1.5, the median day of appearance of antibodies was the 10th, whereas for the group of patients with a medium and high severity of the course of the disease, the median days were 17 and 15, respectively. Thus, the absence or delayed appearance of antibodies to linear epitopes can be associated with a weaker recognition of the viral S protein by the immune system, causing a more severe course of the disease. No antibodies to linear epitopes to other non-structural coronavirus proteins from our panel were observed.

### 3.2. Cross-Reactivity of Antibodies against SARS-CoV-2 with Human Proteins

Using western blot analysis, we observed cross-reactivity of IgG from COVID-19 patients’ serum (Figure 5a,b, Appendix A), mainly for specific to nucleoprotein antibodies. We performed a competitive western blot analysis. In the case of the preincubation of serum with nucleoprotein, we observed a significant decrease of antibody binding with eukaryotic proteins (Figure 5c,d). Serum from healthy donors collected before the pandemic was used as a control. Human proteins were obtained from A549 (lung adenocarcinoma) and T84 (colorectal carcinoma) eukaryotic cell lines, because these tissues are among the main targets of the virus. The binding of antibodies was observed for both cell types, and localization of the bands was identical. Using mass-spectrometry analysis, we identified the major bands (Table 1, raw data are presented in Appendix A). 

A bioinformatic approach was used for comparing the amino acid sequences of N and S SARS-CoV-2 proteins with the sequences of the proteins of cell line A549, T84, and separately considered proteins identified by mass spectrometry (referred to as target proteins, blue squares) (Figure 6, Appendix A). Importantly, most of the overlapping regions for NP were in mutational regions. 

We also performed a bioinformatic analysis of N and S viral proteins for possible presentation of their peptides in HLA I and II classes (Appendix A). The most common HLA alleles were selected for that purpose. According to the results, the HLA-A01:01; HLA-A26:01; HLA-B39:01; and HLA-B15:01 most likely can present viral peptides that are similar to the sequence of human proteins, which can lead to the development of autoimmune reactions. Amino acid sequences, identical to our target proteins (KRT8, KRT18, TUBA1B, ENO1, and EF1A1), were identified in the analyzed peptides. All these proteins could be exposed on the cell surface and cause cross-reactivity of IgG against SARS-CoV-2 with own proteins. No overlapping amino acid sequences were observed in the list of the peptides obtained via the bioinformatic analysis (virtual processing of HLA I and II proteins) between human vimentin and actin, and virus nucleoprotein and spike. 

## 4. Discussion

Historically, viral infections have had a complex relationship with a variety of autoimmune diseases [38], and the new COVID-19 pandemic caused by coronavirus SARS-CoV-2 is not exception [39,40]. There are multiple mechanisms by which viruses can induce an autoimmune reaction [41,42,43], one of which is the cross-reactivity of antibodies secreted against the pathogen to the human’s own proteins. In order to understand the humoral immunity to SARS-CoV-2 and the mechanisms of development of autoimmune processes after COVID-19, we studied the profile of specific IgG antibodies in the sera of patients hospitalized with COVID-19 to linear and conformational SARS-CoV-2 protein epitopes. The analysis was carried out on a sample of 149 patients, and for each of them, sera were collected at several time points to analyze the development of the humoral response throughout the duration of the disease. Sera from healthy donors collected before the commencement of the pandemic were used as controls.

We observed significant differences in the antibody titers starting from the 16th day of the disease only. The highest percentage of detection of IgG antibodies to viral proteins in patients was observed for the nucleocapsid protein, both for conformational and linear epitopes. In addition, according to ELISA data, we found correlations between the titer of antibodies to NP, the severity of the course of the disease (Figure 1a), and complications such as the patient’s oxygen demand (Figure 2). In the high severity group, the antibody titer was 2.5-times higher than in the medium and mild severity groups. Extensive information is currently available about the SARS-CoV-2 virus organization, and the role of antibody to the S protein is widely discussed. However, it remains unclear why antibodies to the nucleocapsid protein are among the major ones, whereas the protein is localized inside the virus and does not play a direct role in penetration. Several lines of evidence were reported suggesting that the SARS-CoV nucleocapsid protein modulates various host cellular processes to make the internal milieu of the host more conducive for the survival of the virus. Moreover, the nucleocapsid protein was detected in serum samples from SARS [44] and SARS-CoV-2 [45] patients. This can be a viable explanation for such a high level of antibodies to this protein. Thus, there are two alternative assumptions about the correlation between a high titer of anti-N protein antibodies and disease severity: (i) higher titers of the antibodies are associated with higher viral loads causing respiratory distress, and (ii) the anti-N antibodies are damaging by themselves. Several studies have already shown [15,16] cross-reactive effects of antibodies from COVID-19 patients’ serum causing autoimmune processes and consequential deterioration of patient condition. However, no such effects were observed for vaccines with the spike protein [46]. Thus, it can be assumed that the development of autoimmune processes is primarily associated with antibodies against NP.

Using chip array, we confirmed our data according to antibodies of conformational epitopes to the structural (N, S) proteins, and we also detected IgG to a non-structural protein NSP5 (Figure 3). Rare immune responses to that protein have been reported before [6]. It is a main virus protease, which targets RIG-I and mitochondrial antiviral signaling (MAVS) protein. In addition, NSP5 potently inhibits interferon (IFN) induction [47,48]. However, antibodies to NSP5 were detected in extremely rare cases; hence, we could not consider the presence of antibodies to NSP5 as a prognostic or diagnostic marker of the disease.

We investigated the putative autoimmunity effect of antibodies using A549 (lung adenocarcinoma) and T84 (colorectal carcinoma) eukaryotic cell lines because these tissues are among the main targets of the virus (Figure 5a,b and Figure 6). Considering that almost 82% of the studied patients had antibodies to the linear epitopes of NP and based on WB data, the observed cross-reactivity of antibodies was likely associated with this protein. We performed a competitive analysis by preincubation of the serum with NP and subsequent western blot analysis with eukaryotic proteins (Figure 5c,d). After such an anti-NP specific antibody elimination, cross-reactivity of the antibodies from the sera with eukaryotic proteins significantly decreased. However, binding of the sera from the control group with cell lysates was also detected. This may be due to the presence of antibodies to other prior coronavirus infections, especially given the high homology among the viral N protein [49], as well as due to possible latent autoimmune processes. The list of bands identified by mass-spectrometry is shown in Table 1. 

Other identified proteins were cytokeratin 18 (CK18) and its co-expressed complementary type II keratin partner CK8, persistently expressed in a variety of adult epithelial organs, such as liver, lung, kidney, pancreas, and gastrointestinal tract [50], which are among the most commonly affected organs in COVID-19. Furthermore, keratin 18 is involved in the uptake of thrombin-antithrombin (TAT) complexes by hepatic cells [51]. The TAT complex reflects the functional state of the coagulation system. Thus, in the presence of autoantibodies to CK18, there may be disturbances in the uptake of this complex, which can lead to abnormalities in the blood clotting process observed in the most severe cases of COVID-19. In addition, thrombin can cross the blood-brain barrier, destroying neurons and potentially causing cerebral edemas [50]. Moreover, it was shown that patients with COVID-19 had higher circulating levels of cleaved CK-18 versus the control [52].

Structural damage to the respiratory epithelium and abnormal ciliary function are the typical pathologic symptoms of CoV-2 infection. Cilia is a composite structure based on microtubules (MTs) presented on the cell surface [53]. In this regard, the presence of autoantibodies to tubulin, which is the main structural element of MT, may explain the occurrence of persistent anosmia. Furthermore, several studies have shown that the disruption of MTs is related to neurodegenerative diseases [54]. MTs combine with motor protein families to take part in long-distance transport in neuronal dendrites and axons [55]. 

Human elongation factor 1-alpha (EF1α), another identified target protein, has unconventional functions by binding to the cytoskeleton [56]. It was shown for SARS-Cov and another viruses interaction between the N protein and EF1α [57,58]. So, it could be the main possible target for cross-reactivity with NP autoantibodies. 

As a result of the study, we found out that, in monitoring the COVID-19 disease progression, it is essential to measure not only the level of antibodies to the Spike protein, but especially to NP (both linear and conformational epitopes) because the main autoreactive antibodies are associated with nucleoproteins. Moreover, for RBD, the appearance of antibodies to the linear epitopes to this antigen correlated with a milder form of COVID-19. People with HLA alleles A01:01; A26:01; B39:01; B15:01 are most susceptible to the development of autoimmune processes after COVID-19. Unfortunately, due to the limitation of our study being the inability to check the presence and titer of neutralizing antibodies in the sera of patients due to the lack of permission to work with the SARS-CoV-2 virus, we are unable to correlate the amount of neutralizing antibodies with the level of cross-reactivity antibodies. The obtained results can be used in clinics to anticipate possible autoimmune consequences after COVID-19, as well as for the further development of vaccines against the coronavirus infection.

## Figures and Tables

**Figure 1 viruses-14-01141-f001:**
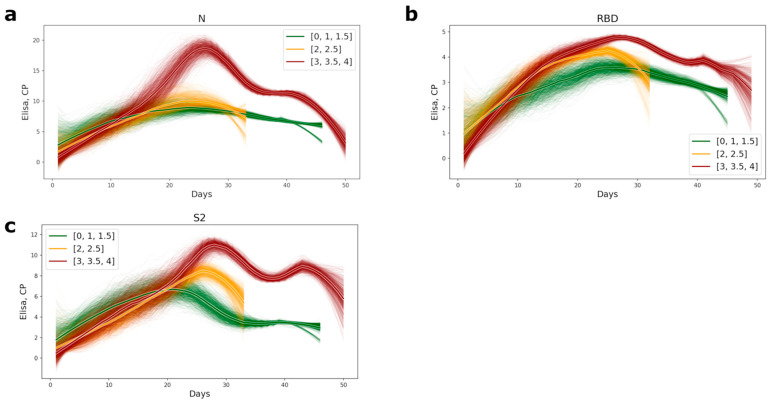
Plots of appearance of the antibodies to the conformational epitopes of N (**a**), RBD (**b**), and S2 (**c**) proteins in COVID-19 patients’ serum. The lines were computed as subsampled sliding windows with a window width of 0 and a subsampling size of 7. The bolder lines are the averages of approximations. The color of the lines represents the level of disease severity: the green lines, the mild course of disease (CT = 0–1.5); the yellow, the medium level of disease (CT = 2–2.5), and red, the most severe (CT = 3–4).

**Figure 2 viruses-14-01141-f002:**
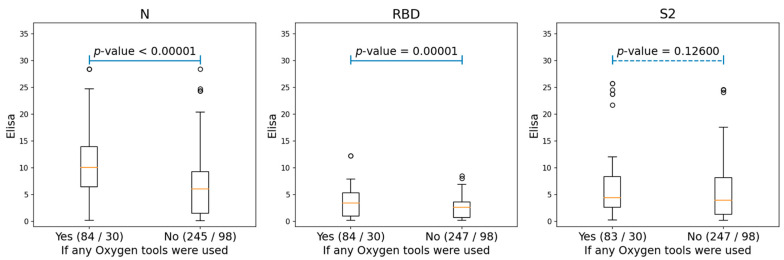
The boxplots, showing the differences for the ELISA values of N, RBD, and S2 proteins among the patients who needed oxygen support and those who did not. The ‘Yes’ group, indicating patients who used any oxygen tools (either nasal cannula or supplementary oxygen) and the ‘No’ group, consisting of patients without any oxygen tool interventions. The two numbers in braces for each group indicated the number of Elisa samples and patients correspondingly. The *p*-values were produced by performing ANOVA tests for each of the two groups.

**Figure 3 viruses-14-01141-f003:**
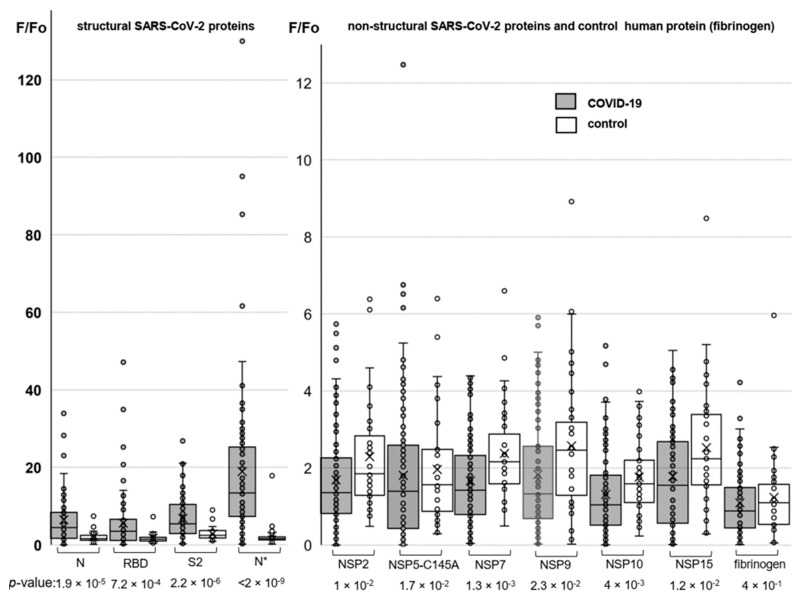
Profiling responses to the structural (left) and non-structural (right) SARS-CoV-2 proteins using protein microarrays. The box plots summarize the distributions of the relative levels of antibodies to particular proteins detected in COVID-19-positive donors (numbers of samples: 95 for SARS-CoV-2 proteins and 80 for fibrinogen) and the control group (numbers of samples: 37 for SARS-CoV-2 proteins and 34 for fibrinogen).

**Figure 4 viruses-14-01141-f004:**
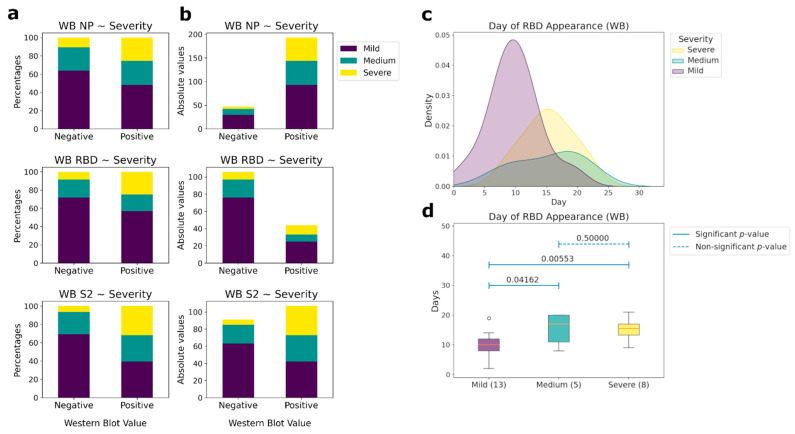
Correlations between the presence of IgG antibodies to linear epitopes NP, RBD, and S2 and patient’s condition severity (western blot (WB) data). (**a**) Comparative analysis between the presence (positive) or absence (negative) of antibodies and the severity of patient condition according to CT in percentages. (**b**) Comparative analysis between the presence (positive) or absence (negative) of antibodies and the severity of the patient’s condition according to CT in absolute values. (**c**) Histograms of the day of appearance of antibodies for those samples, for which the day of the beginning of the disease is known, to the linear (western blot) epitopes of RBD, depending on the severity of patient condition. (**d**) Comparative analysis of the day of the antibodies presence to the linear epitopes of the RBD protein and the severity of the patient condition for those samples, for which the day of the beginning of the disease is known. The number of patients in each severity group, for which the linear epitopes were measured, is outlined in brackets. Pairwise comparisons were performed using the Mann-Whitney test. *p*-values of the corresponding comparisons are presented above the blue segments. Solid segments correspond to the significant *p*-values (≤0.05), dashed segments represent non-significant *p*-values (>0.05).

**Figure 5 viruses-14-01141-f005:**
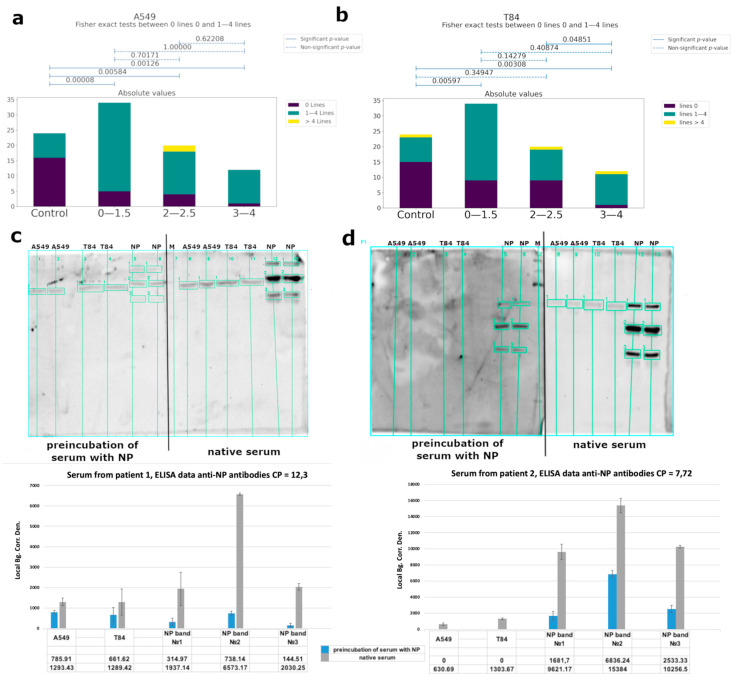
Analysis of the cross-reactive binding of IgG antibodies from the sera of patients with COVID-19 with linear epitopes of human proteins. Serum from healthy donors obtained before COVID-19 time was used as a control. (**a**,**b**) Correlations between the number of bands recognized by antibodies and groups of patients of different severity (according to CT data) for A549 cells and T84, respectively. (**c**,**d**) Analysis of the cross-reactivity of IgG from the serum before and after the separation of anti-NP specific antibodies from two different patients’ serum, respectively, with eukaryotic proteins by western blot and quantitative values of bands chemiluminescence calculated using the built-in software iBright Imaging Systems (ThermoFisher). The table shows the average signal intensity values. Local Bg. Corr. Den.—Local Background Correction Density—The Local Background Corrected Volume divided by the Area.

**Figure 6 viruses-14-01141-f006:**
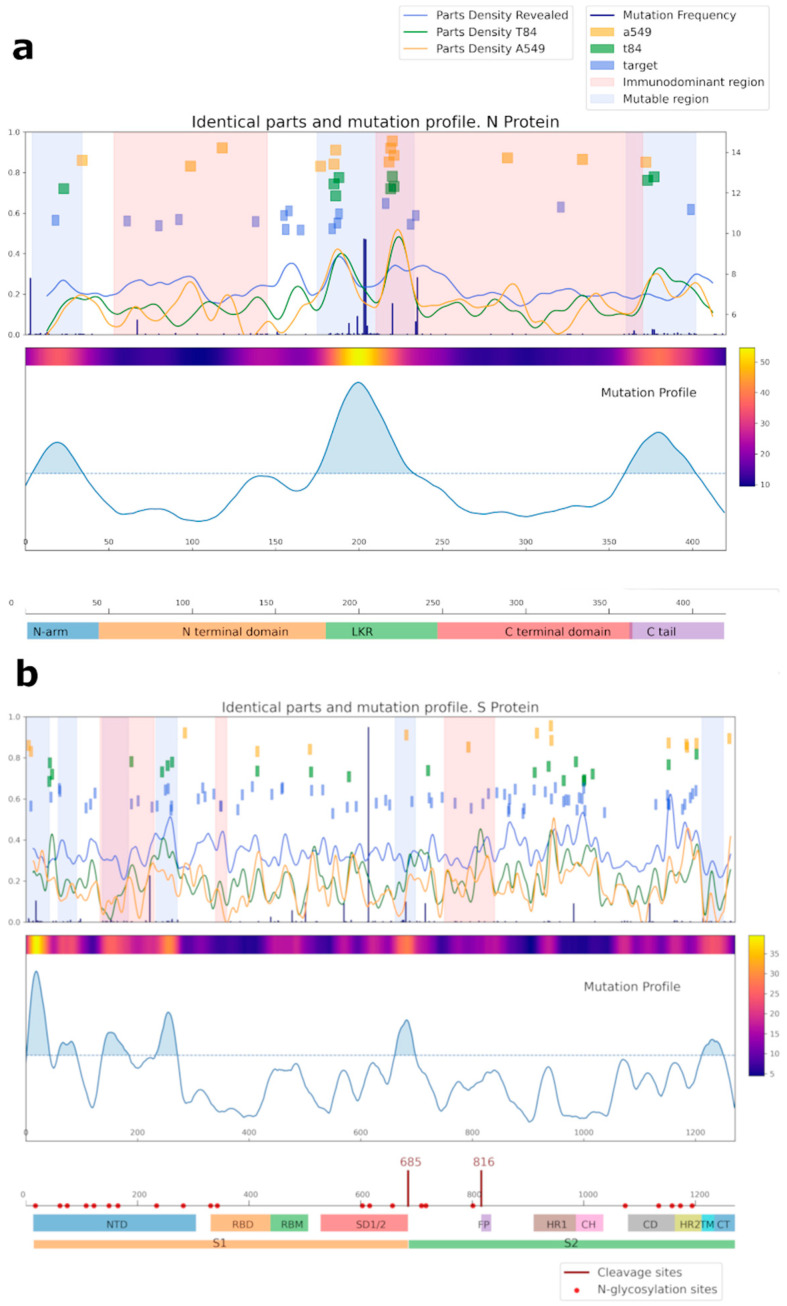
Analysis of the common regions between N (**a**), S (**b**) SARS-CoV-2, and human proteins. Proteins from the A549 cell line are shown in yellow; proteins from the T84 cell line are in green; our target protein, detected by mass-spectrometry as potentially capable of being recognized by IgG antibodies, is in blue. For cell lines, the intersection area was selected at 6 amino acids; for target proteins, it was selected at 4. Mutational and immunodominant profiles of proteins were taken from published data.

**Table 1 viruses-14-01141-t001:** List of the identified proteins from eukaryotic cell lysates, which were recognized by antibodies from the COVID-19 patient sera.

Accession Number	Description	Score	Protein Mass, Da	Peptide Matches	Protein Coverage, %
**T84 cell line**					
K1C18_HUMAN	Keratin, type I cytoskeletal 18 OS = Homo sapiens GN = KRT18 PE = 1 SV = 2	190	48,029	23	59.1
ACTG_HUMAN	Actin, cytoplasmic 2 OS = Homo sapiens GN = ACTG1 PE = 1 SV = 1	152	41,766	19	55.3
K2C8_HUMAN	Keratin, type II cytoskeletal 8 OS = Homo sapiens GN = KRT8 PE = 1 SV = 7	146	53,671	23	58.2
VIME_HUMAN	Vimentin OS = Homo sapiens GN = VIM PE = 1 SV = 4	88	53,619	17	45.1
EF1A1_HUMAN	Elongation factor 1-alpha 1 OS = Homo sapiens GN = EEF1A1 PE = 1 SV = 1	43	50,109	10	25.1
**A549 cell line**					
VIME_HUMAN	Vimentin OS = Homo sapiens GN = VIM PE = 1 SV = 4	96	53,619	17	42.5
TUBA1B_HUMAN	Tubulin alpha-1B chain OS = Homo sapiens GN = c PE = 1 SV = 1	64	50,120	13	41.5
K1C18_HUMAN	Keratin, type I cytoskeletal 18 OS = Homo sapiens GN = KRT18 PE = 1 SV = 2	211	48,029	24	59.3
ACTG_HUMAN	Actin, cytoplasmic 2 OS = Homo sapiens GN = ACTG1 PE = 1 SV = 1	67	41,766	12	43.2
ENOA_HUMAN	Alpha-enolase OS = Homo sapiens GN = ENO1 PE = 1 SV = 2	84	47,139	15	49

## Data Availability

Clinical data files are stored at Scientific Research Institute of Systems Biology and Medicine. It may be shared if need.

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
