# Peer review of "Autoimmune Effect of Antibodies against the SARS-CoV-2 Nucleoprotein"

_viruses, 2022, doi:10.3390/v14061141_

Round 1

Reviewer 1 Report

In the present manuscript by Matyushkina et al., a thorough analysis of the different antibodies against COVID19 and their various correlations are presented. In addition, a cross-reactivity analysis was also performed in an attempt to define part of the mechanisms governing the generation of autoimmunity following a Covid 19 infection. Even though the manuscript is well written and the scientific question solid, there are some clarifications needed before meriting publication.

  • It would be nice if you could rephrase the first sentence of the introduction. If you consider it necessary, please specify the acute respiratory diseases associated with coronaviruses.
  • Please add suitable references in the first paragraph of the introduction.
  • To my mind stating that today no sufficiently effective antiviral drugs are available for COVID-19 is an exaggeration if not wrong. Recently paxlovid showed an over 80% death risk reduction.
  • In my opinion, the statement "statistics indicate that SARS-CoV-2, despite its increased contagiousness, has been causing fewer deaths than the previous two representatives of coronaviruses" could be amended.
  • It would be also interesting to include in the "introduction" part a brief overview of the autoimmune sequela of a COVID-19 infection.
  • ADE (antibody-dependent enhancement) that can occur after infection or vaccination, is not an autoimmune phenomenon, the few, not well-substantiated reports have been related only to vaccinations and not natural infections regarding SARS-COV-2 and in my opinion is out of the scope of this study.
  • Grading the severity of the COVID-19 infection in each patient, based on the CT findings is controversial and in my opinion wrong. On several occasions, there is a great discordance between the radiographic findings and the clinical picture of the patient. The use of a score that reflects in a better way the clinical status of the patient such as the PaO2/FiO2 ratio may be more appropriate to cluster your patients into groups. If you choose not to change the stratification of your patients always specify that the patients are grouped based on the RADIOGRAPHIC severity of their disease.
  • What do you mean by disease inception?
  • Could you please specify the number of days between the first symptom of the disease and S1(admission to the hospital) and compare this time period in the different groups. Moreover, a great disparity in the time between the onset of symptoms and S1 can also affect the accuracy of what you considered a high antibody level.
  • Please create a table with information on the number of patients allocated in each severity group, the group median score, the median age of the patients, and the most important clinical information. In addition, did any of the patients die because of COVID-19? Were samples for S1-4 available for all patients? Is censoring a limitation of the study?
  • In my opinion and in an attempt to facilitate the reader figure 2 could be simplified in just the boxplots. All the other correlations performed, even though not revealing a statistical significance, could be shown in the supplementary.  
  • Please use the full word, post-translational modification instead of PTM.
  • Please specify which patient's sample (S1-4) was used in the protein microarrays assay.
  • In figure 3 please add the p-value of all the different comparisons made.
  • In figure 4 panel a, showing just the percentages and not the actual numbers could lead to false deductions.
  • In the legend of figure 6 please fill-up the color blue that is missing from describing the target protein 
  • Please add a paragraph with the limitations of the present study
  • Personally I think that the conclusions should be rephrased in a more modest and unassertive way 

Reviewer 2 Report

Comments:

- A hyphen should not be used in the title. It is in the word nu-cleoprotein. I recommend moving the word lower or using the abbreviation - NCP. Then, in the abstract and in the introduction, this abbreviation should be explained.

- In Materials and Methods - "Study Population" it is not enough to give the median age, because it does not give information about the age structure of the population. Its characteristics are indicated.

- In the subsection "ELISA and CMIA" the units "mkl" are given. They are probably microliters. Please use the generally accepted and used unit - "µl"

- Line 131/132 - unnecessary signs "« ... »". If they are justified, please provide arguments.

- Line 138/139 - critical and mean: "ODcritical = ODmean K + 0.2" and "ODmean". They should be subscript.

- Line 140/141 - subscript: "CP = ODsample / ODcritical" It is unclear whether the CP reference ranges apply to both methods. No reference range of results for the Abbott "SARS-CoV-2 IgG II Quant assay" assay.

- Line 156 - there is an incorrect abbreviation for "mkg". Suggests that milli Kg was used. Please correct.

- Line 208 - remove unnecessary word "Statistics"

- In the chapter "Statistics" (line 209) the description of the methodology used is too general and makes it impossible to assess the reliability of the results obtained. The software that was used for each calculation was not even reported, and no statistical assumptions were made for the individual methods.

- Figures 2 and 4 are of poor quality and have a vague description.

- Figures 1, 2, 5 and 6 are too small letters.

- For Figure 2, there is no statistical confirmation for the conclusions.

- Figure 5 caption is about naive serum or native serum?

- Due to the subject matter, the discussion should more thoroughly refer to the available data related to autoimmune processes in COVID.

- For "Authors' contributions:", entries such as "Conceptualization,", "Data curation," and so on, should have a colon.

Round 2

Reviewer 1 Report

Most of my comments have been resolved by the authors.